# Effectiveness of a Computer-Based Training Program of Attention and Memory in Patients with Acquired Brain Damage

**DOI:** 10.3390/bs8010004

**Published:** 2017-12-30

**Authors:** Elizabeth Fernandez, Jorge A. Bergado Rosado, Daymi Rodriguez Perez, Sonia Salazar Santana, Maydane Torres Aguilar, Maria Luisa Bringas

**Affiliations:** 1Neuropsychology Department, International Center for Neurological Restoration CIREN, 11300 Havana, Cuba; fernandezelo43@gmail.com (E.F.); sonia@neuro.ciren.cu (S.S.S.); mtorres@neuro.ciren.cu (M.T.A.); 2Laboratory of Electrophysiology, International Center for Neurological Restoration CIREN, 11300 Havana, Cuba; jorge.bergado@infomed.sld.cu; 3Department of Psychology, Medical and Surgical Research Center CIMEQ, 11300 Havana, Cuba; dayrp@infomed.sld.cu; 4The Clinical Hospital of Chengdu Brain Science Institute, MOE Key Lab for Neuroinformation, University of Electronic Science and Technology of China, Chengdu 610054, China

**Keywords:** cognitive rehabilitation, computer assisted therapy, acquired brain damage, attention, memory, RehaCom

## Abstract

Many training programs have been designed using modern software to restore the impaired cognitive functions in patients with acquired brain damage (ABD). The objective of this study was to evaluate the effectiveness of a computer-based training program of attention and memory in patients with ABD, using a two-armed parallel group design, where the experimental group (*n* = 50) received cognitive stimulation using RehaCom software, and the control group (*n* = 30) received the standard cognitive stimulation (non-computerized) for eight weeks. In order to assess the possible cognitive changes after the treatment, a post-pre experimental design was employed using the following neuropsychological tests: Wechsler Memory Scale (WMS) and Trail Making test A and B. The effectiveness of the training procedure was statistically significant (*p* < 0.05) when it established the comparison between the performance in these scales, before and after the training period, in each patient and between the two groups. The training group had statistically significant (*p* < 0.001) changes in focused attention (Trail A), two subtests (digit span and logical memory), and the overall score of WMS. Finally, we discuss the advantages of computerized training rehabilitation and further directions of this line of work.

## 1. Introduction

Acquired brain damage (ABD) is brain damage that occurs after birth and is not related to a congenital or degenerative disease. These impairments may be temporary or permanent and cause partial or functional disability or psychosocial maladjustment. (Definition and more related information at http://braininjurysociety.com/information/acquired-brain-injury/what-is-abi/). ABD is a frequent cause of alteration in cognitive function, communication capacity, and abilities to regulate behavior, because of the high incidence rate and the long duration of the effects, having a negative impact on life quality that relates directly with the degree of cognitive dysfunction. This is closely related with the loss of autonomy that constitutes the most important concern for patients and caregivers [1,2]. 

Major causes of ABD are stroke and brain traumatic injury [3,4,5]. Both conditions share varied, but in many instances, similar, neuropsychological impairments: attention deficits, impaired memory, impaired executive functions, apraxia, alexia, agraphia, aphasia, among others, which depend on the extension, severity, and location of the injury [6,7,8,9,10]. These neuropsychological sequels impair the social integration of patients even more than the physical limitations [11,12,13,14]. For that reason, the interest in cognitive rehabilitation has been growing over the last decades [11,15,16,17]. Cognitive rehabilitation can be defined as the set of procedures and techniques that are aimed to improve intellectual efficiency, and the adaptation to familiar, social, and labor environments of patients suffering brain damage [17,18]. 

Neuropsychology has, consequently, broadened its field of application, both in the clinic and research, to design therapeutic strategies for the rehabilitation of cognitive disorders, based on neural plasticity as the biological substrate of recovery [19,20]. 

On the other hand, the use and combination of the new functional neuroimaging techniques gives the opportunity to register the reorganization of the brain with a high temporal and spatial resolution, providing objective measures to assess the effectiveness of the rehabilitation programs. A review about neuropsychological rehabilitation results correlates with functional neuroimaging in patients with brain damage in [21]. 

An important number of programs for neuropsychological rehabilitation have also benefited from developments in Neurotechnology. In this sense, multimedia systems have been implemented to create interactive tools that facilitate the interaction with the patient and the treatment of cognitive dysfunction in persons suffering sequels from ABD [11,22,23,24,25]. Reports on the use of such programs are highly variable regarding the techniques employed, therapy duration, intensity of treatment, study design, and the tools to evaluate their efficacy. Reviews on the subject have emphasized on the need of high quality studies to assess the efficacy of cognitive rehabilitation programs using objective measures to facilitate the interpretation of their real usefulness [14,26]. 

Our study was aimed to provide evidences about the effectiveness of a computer-based training program of attention and memory in patients with ABD, using a two-armed parallel group design, in comparison with a standard (non-computerized) cognitive training program, expecting that the group who received computerized training have higher cognitive performance after the intervention. 

In addition, we explore the possible relationship between cognitive recovery and other clinical and demographic variables like age, time of evolution (time since the onset of the acquired brain damage), and premorbid educational level before treatment. 

## 2. Materials and Methods

### 2.1. Sample

The study was a two-arm parallel group design, where the experimental group (*n* = 50) received cognitive stimulation using RehaCom software, and the control group (*n* = 30) received the standard cognitive (non-computerized) stimulation. 

For the selection of the subjects we employed the convenience or availability sampling method that relies on data collection from population members who are conveniently available to participate in the study. In this case, the population consisted in the adult patients that were suffering cognitive dysfunction after ABD admitted consecutively at the Static Lesion Clinic at CIREN (www.ciren.cu) for neuro-rehabilitation treatment between 2010 and 2014. The total number was 312 and 80 patients were eligible (see below exclusion criteria) to be included in this protocol after giving their informed consent. 

The both groups received a comprehensive and intensive neurorestoration program designed at CIREN for cognitive, motor, and physical rehabilitation during eight weeks, according to the Intensive Multifactorial Rehabilitation Program, which is in use in this institution. The program included 2 h daily of cognitive rehabilitation. As explained before, 50 received cognitive specific training using RehaCom software (www.schuhfried.at) to constitute the “training group”. The remaining 30 received the conventional cognitive rehabilitation using a program of activities (non-computerized) to stimulate attention, memory, and executive function, the “control group”. The subjects who fill the inclusion criteria were allocated to the different groups, according to the availability of space at the computer room. Exclusion criteria were: severe cognitive dysfunction, aphasia, illiteracy, degenerative diseases, or mental retardation before ABD were not included in either group. All of the work was conducted in accordance with the Declaration of Helsinki (1964). The demographic information of both samples is shown in Table 1. 

### 2.2. Methods 

The Mini Mental State (MMSE) examination was applied initially to all of the participants to evaluate their initial cognitive condition and confirm the equivalence between groups at the baseline [27]. To evaluate the effectiveness of the intervention two neuropsychological tests were used, the Trail making Test (TMT) (Form A and B) and Wechsler Memory Scale (WMS) [28,29]. Note that the evaluations were performed using a pencil and paper standard assessment. The raw results of each test and subtest were compared within the group before and after training, using a paired sample *t*-test. To compare results between groups, we performed a repeated measures ANCOVA for fixed effects, where we analysed the differential scores of attention and memory tests between the two times (pre and post training). This differential score *c* was calculated using a formula *log10(post+1)/log10(pre+1)* to overcome the differences between the outputs of the different scales employed. 

We included in the model an adjustment for age, education level, and time of evolution (as covariates), and their interaction with “group”, fitting a linear model for *p* = 11 variables (model ≤ lm(Y[[p]] ~ group + age + evolution + schooling + group*age + group*schooling + group*evolution) and calculating the ANCOVA (using the functions *lm* and *anova* from the R package “stats” version 3.3.3.). 

The Pearson coefficient of correlation between the initial MMSE and the neuropsychological measures was employed. 

### 2.3. Rehabillitation Procedures

For the experimental group, we employed the software RehaCom, which is a modular, interactive program that is designed to train cognitive abilities. The system design includes compensatory strategies, controlled stimuli, and immediate feedback. The system includes procedures to train and improve attention, memory, visuo-spatial processing, and executive functions www.schuhfried.at [30]. The therapist’s interface allows for the introduction and retrieval of personal and clinical information of the patients, the design of individual subprograms, including the individualized level of difficulty (or baseline), along with the collection of data. The therapist can also execute the tests with the patient using the demo version before the definitive inclusion in the rehabilitation training. 

The training session consists in the performance by the patients of the subprograms that were selected by the therapist at the pre-established degree of difficulty. The system computed the individual results including reaction time, scores and number of errors. 

The sessions were divided into five 50-min sessions per week. All of the sessions were held in the morning in a laboratory designed for this purpose, where up to four patients worked simultaneously under supervision by two specialists. Patients were monitored for possible negative effects during training.

### 2.4. Procedure of the Intervention

The initial cognitive condition was first established, as well as the patient’s self-awareness of the deficits as premises to organize the sequence of the sessions of rehabilitation. To rehabilitate the processes of attention and memory, a restorative strategy was implemented using repetitive exercise, increasing the level of difficulty after successful performance of the task. Within each task, an immediate feed-back educated the patient about the results of the execution. At the end of each session, a graph was provided to show to the patient the results obtained in every task.

To implement the training of the attention, we follow the proposals of Sohlberg and Mateer starting with exercises that stimulate the level of activation, processing speed, focal attention; to work later sustained attention, alternating attention, and finally, executive attention [31].

To train attention, behavior/reaction time procedures were implemented to improve the velocity of information processing. The subject was prompted to maintain the activation level waiting for the presentation of an imminent stimulus associated with a specific response. While complexity increases, the executive components of attention are trained; i.e., the ability to displace attention from one set of attributes to a different one. This capacity is essential for cognitive flexibility. Detection and response to stimuli arriving from the left or right visual field was also trained.

Attention and concentration procedures were used to train attention stability, maintenance of the vigilance focus, and the ability to compare patterns. It also included the training in tasks of visual search. Subjects had to search on a screen for an object, previously defined as target, and discriminate it from simultaneous distractors. In exploration and vigilance tests, the detection of low frequency stimulus in monotone and prolonged context was also trained.

The training of memory functions was organized in successive phases: basic attention, information coding, consolidation, and retrieval. Compensatory strategies were introduced to facilitate the understanding, organization, and categorization of the items. Procedures for memory of objects, faces, and text were included as well, supported by facilitation and recognition strategies. 

Integrative procedures to stimulate executive and generalization functions, logical reasoning, calculation, numerical reasoning, and other that simulate daily life situations (i.e., simulated shopping) were also applied.

The control group received a cognitive stimulation based in the same principles mentioned above, consisting in a set of activities that are aimed to activate cognition, attention, concentration, visuo-constructive abilities, memory, and executive functions using cards, paper-and-pencil tasks, which were combined with other activities to train fine motor functions. The main strategies employed to improve coding and retrieval were repetition, centralization, and elaboration. 

## 3. Results

All patients included in the sample completed the treatment scheduled in eight weeks. The Table 2 included the mean and standard errors for each neuropsychological variable before and after intervention and the within group significant differences. All of the individual scores, including demographic and clinical information, can be accessed as a supplementary material (Bringas Fernandez NP 2016 excel sheet). 

### 3.1. Analysis Within Groups

The within student’s *t*-test (paired samples, *p* < 0.05) showed significant improvements in both of the treatment groups. The improvement was also statistically significant in all of the subtests of the memory scale, as well as the velocity of information processing, focused and executive attention in the Trail Making Test (TMT) (forms A and B). Note that the efficiency in the TMT is expressed in seconds, therefore faster results (lower scores) are better. The opposite applies for the Wechsler Memory Scales, where the higher scores indicate better performance. 

### 3.2. Analysis between Groups

The between groups comparison (before and after treatment) using an ANCOVA with the differential scores of the neuropsychological tests and age, evolution time, and educational level as covariates is showed in Table 3. 

The summary of the findings are: (a)The age did not show any effect on neuropsychological variables and the rest of the covariates.(b)The main effect of the *group* was related to four variables: the focused attention of the TMT (Trail A) (*p* = 7.698 × 10^−606^), two subtests of the WMS, digit span (*p* = 0.008) and logic memory (*p* = 0.01) and the Memory Quotient (*p* = 1.118 × 10^−5^). (c)The main effect of *time of evolution* was related to subtests orientation (*p* = 0.025) and associative learning (*p* = 0.010) of the WMS. Patients with more time of evolution of the ABD had the worst performance in these subtests. (d)The main effect of the *educational level* with WMS subtest orientation (*p* = 0.001). (e)The only interaction we found between *group* and the covariates was in the associative learning, where not main effect of group was found, but the interaction with *educational level* (*p* = 0.049) and *time of evolution* (*p* = 0.024*) was statistical significant. 

Figure 1 presents the results of the between groups comparison of the performance before and after intervention. The results showed a significantly higher improvement in attention (focused and/or sustained) for the groups that received computer aided training when compared to the control group. The executive attention did not show statistical significant differences between groups. 

The differential performance of patients in both treatment groups in memory functions, using memory quotient (MQ) is also shown in Figure 2. 

An additional correlational analysis found a significant negative relationship between the initial assessment of global cognitive impairment (MMSE) and the focused and/or selective attention (TMT A, *r* = −0.597), as shown in Figure 3. 

## 4. Discussion

Cognitive rehabilitation has become a necessary intervention in the treatment of sequels of ABD. The results of this study suggest that specific techniques that are employed in the cognitive rehabilitation treatment programs, both conventional and computer-aided, can significantly modify the main cognitive and behavioral impairments of the affected persons. 

All of the patients that were included in the study improved their cognitive performance when compared with their initial condition. These results show that cognitive rehabilitation, even using conventional techniques, can do a significant contribution to improve the neuropsychological performance of ABD patients. On the other hand, the patients of the training group reached a better performance in attention (focused), and memory (digit span, logic memory, and general memory quotient), demonstrating the superiority of the computerized cognitive training in comparison with the non-computerized program. 

### 4.1. Mechanism Underlying the Cognitive Rehabilitation 

The most common approaches to ABD cognitive remediation focus on teaching compensatory strategies to minimize the functional effect of cognitive impairments. Another approach to cognitive remediation aims to restore impaired functions using repetitive exercises or massed practice of specific tasks [16,32]. The repetitive presentation of stimuli and the performance of a response can induce neural plasticity processes that sustain the observed improvements; for example, MRI is able to detect macro-and microstructural activity-related changes in the brain following intensive training [33].

Cognitive functions, especially attention and memory, are important mental tools to adapt the individuals to change. It has been demonstrated that experience induces changes within the Nervous System that modify behavior and may be long-lasting. These neuroplastic properties sustain the efficacy of repetitive stimulation to promote the compensation and recovery of cognitive functions that are affected by ABD [19,20]. 

### 4.2. Benefits of Computerized Rehabilitation Techniques

The introduction of new technologies in Neuropsychology has greatly contributed to increase the success rate of neuropsychological interventions, particularly in the field of rehabilitation [34] in different neurological diseases. One study on patients with multiple sclerosis (recurrent remission and low levels of disability), using a computer-based intensive training program of attention, information processing, and executive function, was effective and also leads to improvement in depression [35]. Reports in dementia and brain traumatic injury respectively using computer-aided cognitive rehabilitation demonstrated improvements in short term visual memory, indicated by [14,30,36]. 

Other example of computer-aided cognitive rehabilitation in stroke patients point out the importance of an improved attention as a fundamental aspect to promote the recovery and transfer to daily life activities. Attention was one of the most favoured areas and the improvement might be transferred to other functional areas related with these processes [10,11,14,30,37].

The application of computerized cognitive rehabilitation also improved cognitive function in elderly persons, the memory test showed increased and significantly different pre and post intervention [38].

Current studies on cognitive rehabilitation in Parkinson Disease demonstrate the improved performance on attention and executive functioning abilities [39].

Summarizing, the implementation of computer aided systems introduced new tools that allow a rapid, flexible, and economic approach for restorative interventions, mainly because is possible to incorporate a greater number of patients to treatment in a motivational environment, which is almost independent of the motor and/or sensory limitations that are imposed by ABD. No adverse events occurred in the present study and most participants reported no significant technical problems. The incorporation of immediate feedback favours meta-cognitive training and a better self-perception of the deficit, promoting the patient’s effort to reach the objectives and a significant improvement in the results. Also, the computer allows for a higher level of stimulation and activation, improves the quality of the stimuli and their presentation. This can, by itself, improve attention and focus at a multisensory level (i.e., images combined with sounds). Computer systems also allow better control and correction of failures by a constant feed-back to the patient as our results indicate [11,25,32,33]. Those studies have shown that memory functions benefit after specific training to develop mnemonic abilities. In correspondence with these findings, we report significant improvements in visuo-spatial memory and verbal memory in a sample of ABD patients.

Executive attention supposes a more complex process, and our results indicate that a longer training time is required to obtain significant benefits in these useful mechanisms, even for daily life functions [3,16]. Computer-assisted programs allow for memory recovery, independently of the initial degree of dysfunction. These programs are flexible and allow for a constant adjustment of complexity of the tasks according to the needs of each patient, provide feed-back and reinforcement that increase the motivation of the patient.

### 4.3. Global Cognitive Functioning

Another aspect to be considered is the global cognitive functioning of the neurologic patient, which can be evaluated using the MMSE. The integrity of cognitive processes facilitates the appropriation of the strategy and of the therapeutic procedures that are aimed to induce the recovery of affected functions. A better cognitive condition could also be related with the self-awareness of the deficit. Therefore, the best preserved patients have a better performance and a higher motivation in the tasks. Independent studies in normal subjects have demonstrated a strong relationship between the speed of information processing and cognition, suggesting that the learning is more efficient when the person makes a better use of its cognitive resources [33]. An additional correlational analysis found a significant negative relationship between the initial assessment of global cognitive impairment (MMSE) and the focused and/or selective attention (TMT A, *r* = −0.597). Patients with higher MMSE scores at the initial evaluation showed better performance in the neuropsychological test, after the intervention. 

### 4.4. Cognitive Recovery and Other Clinical and Demographic Variables

The age did not show any effect on neuropsychological variables and the rest of the covariates, this is consistent because at the initial evaluation, the groups did not show differences at age. Mean and standard deviation training group (33.7 years; 1.77) versus (34.2 years; 2.27). 

Time of evolution and time since the brain injury had an effect (independent of the group) on the spatial and temporal orientation of the subjects. Also, for the associative learning subtest, one of the most difficult memory task because include related and not related pair of words. In both subtests, patients with more time of evolution of the ABD had worst performance, indicating a deterioration process across time. 

The educational level is considered as one of the variables influencing cognitive performance, however our results show no significant relationship between this and the recovery from attention and memory impairments. We only found an independent effect of education level on the orientation subtest. 

On the other hand, we found an interesting statistical significant interaction between group and educational level and time of evolution in the subtest of associative learning. This indicates that for the training group, when the subject has higher educational level and less time of evolution, the performance is much better than the control group. In other words, it is not enough to receive computerized cognitive rehabilitation to improve the performance in this subtest if other covariates are not associated.

### 4.5. Limitations of the Study

There are several directions in which this study can be improved. A long-term follow-up would be very useful to see if the improvements were persistent in time. Though this was impossible in this sample due to patient’s mobility, this is a factor to take into account for future work. Additionally, we were limited by the fact that these were inpatients and we were not able to fully evaluate many aspects of daily living and functional independence. The subjects, being in a hospitalized facility, were constantly attended by nurses and accompanying persons. It would also be useful to gauge how attention and memory functions impact on more complex cognitive functions, such as thought and language—with the aid of psychophysiological and neuroimaging measures. Despite these limitations, we consider that our results might encourage new studies to complement the findings of this report. 

## 5. Conclusions

The intervention “training” group showed significantly higher improvement in attention (focused and/or sustained) and memory (digit span, logic memory, and general memory quotient), demonstrating the superiority of the computerized cognitive training in comparison with the non-computerized program. 

Nonetheless, the two groups, irrespective of receiving conventional or computerized training, improved their general cognitive performance, as measured by standardized neuropsychological tests in patients with ABD. 

## Figures and Tables

**Figure 1 behavsci-08-00004-f001:**
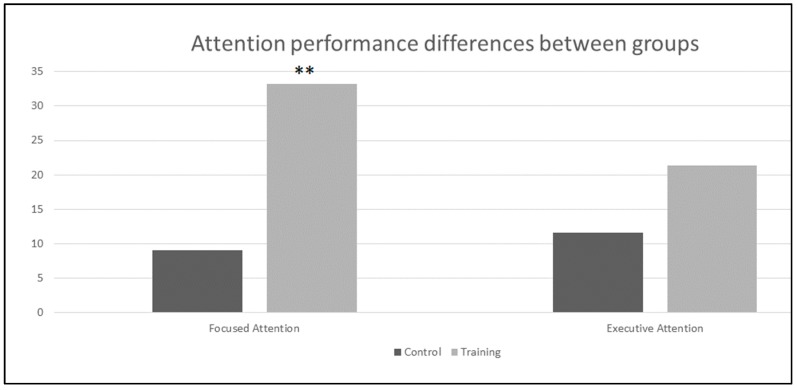
The difference scores at the attention performance. Note: ** indicates the statistical significance between within groups comparison for *p* < 0.05.

**Figure 2 behavsci-08-00004-f002:**
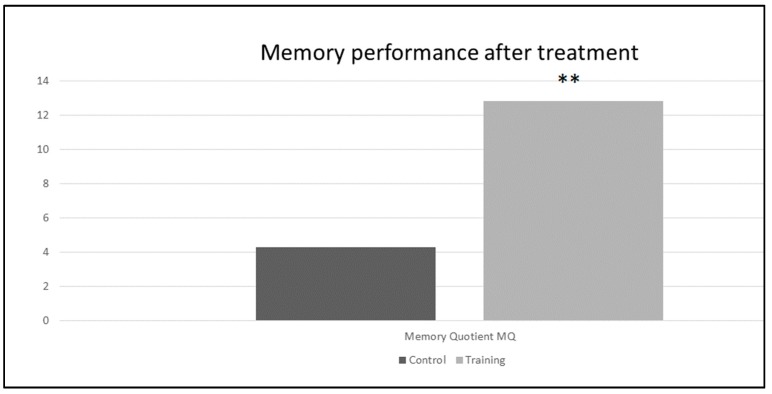
The difference scores in the Memory Quotient after treatment. Note: ** indicates the statistical significance between within groups comparison for *p* < 0.05.

**Figure 3 behavsci-08-00004-f003:**
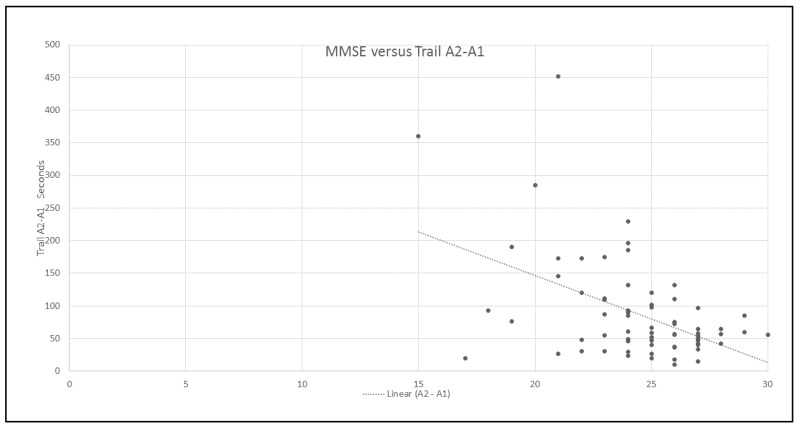
The Mini Mental State (MMSE) scores versus focused attention differences.

**Table 1 behavsci-08-00004-t001:** Description of the sample included in the study.

		Training Group *N* = 50		Control Group *N* = 30	
Gender		No.	%	No.	%
	Male	36	72	18	60
	Female	14	28	12	40
	Total	50	100	30	100
Etiology	TBI	31	62	16	53
	stroke	19	38	14	47
	Total	50	100	30	100
		**Media**	**SEM**	**Media**	**SEM**
	Evolution time	5.6	0.96	5	0.87
	Educational level	13	0.47	13	0.60
	Age	33.7	1,77	34.2	2.27
Initial neuropsychological evaluation	Global (MMSE)	23.4	0.46	25.1	0.30
	Focused attention (seconds)	112	10.9	147.8	8.5
	Executive attention (seconds)	112	15.8	91	11.9
	Memory MQ	75.9	2.24	78,3	2.7

SEM: standard error of the mean; Evolution time: time since the onset of the acquired brain injury until the moment of the evaluation; Educational level: number of years at the school; TBI: traumatic brain injury; Memory MQ: Memory Quotient.

**Table 2 behavsci-08-00004-t002:** Neuropsychological assessment scores pre- and post-intervention.

Variable	Control Group *N* = 30				Training Group *N* = 50			
	Initial		Final		Initial		Final	
	Mean	SE	Mean	SE	Mean	SE	Mean	SE
Focused attention Trail A	147	8.5	138 *	8.13	112.4	10.9	88.5 *	9.35
Executive attention Trail B	91	11.9	80 *	10.17	112.64	15.80	93.1 *	13.82
Orientation	4.2	0.24	4.5 *	0.17	3.98	0.21	4.60 *	0.11
Information	4.1	0.25	4.8 *	0.18	4.50	0.19	5.14	0.14
Mental control	5.9	0.51	6.4 *	0.40	5.70	0.34	7.02 *	0.29
Digit span	7.2	0.33	7.4 *	0.30	7.12	0.27	8.36 *	0.27
Associative learning	9.8	0.69	10.4 *	0.68	9.54	0.57	11.2 *	0.62
Logical memory	6.2	0.62	7.4 *	0.59	5.66	0.50	8.40 *	0.55
Visual memory	6.4	0.59	7.4 *	0.58	6.28	0.40	8.00 *	0.44
Memory Quotient: MQ	78.3	2.74	82.6 *	2.87	75.9	2.24	88.7 *	2.72

Note: * indicates the statistical significance between within groups comparison for *p* < 0.05.

**Table 3 behavsci-08-00004-t003:** Results of the comparison between groups using ANCOVA (fixed effects). Main effects of the Group differences between the differential performance before and after intervention in both groups. Effects of age, time of evolution and educational level and their interaction with group.

Variables	Group		Time of Evolution		Educational Level		Interactions
	F	*p*	F	*p*		*p*	
Focused attention Trail A	23.4	7.698–606 ***					
Orientation			5.24	0.025 *	10.5	0.001 **	
Information							
Mental control							
Digit span	7.31	0.008 **					
Associative learning			6.87	0.010 *			Group*Educational level F = 4.0, *p* = 0.049 * Group*Time of evolution F = 5.24, *p* = 0.024 *
Logical memory	6.97	0.01 *					
Visual memory							
Memory Quotient: MQ	22.35	1.118 × 10^−5^ ***					

Significant probabilities values = * *p* < 0.01, ** *p* < 0.001, *** *p* < 0.0001.

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
