# Peer review of "Effectiveness of a Computer-Based Training Program of Attention and Memory in Patients with Acquired Brain Damage"

_behavsci, 2017, doi:10.3390/bs8010004_

Round 1

Reviewer 1 Report

The manuscript adds important information on rehabilitation after brain damage. However, before publishing, some modifications should be made:

The analysis compare the RehaCom software with a control training. The introduction must therefore emphasize that they are planning to compare the effectiveness of one training over the other. Further, it would be good if the authors outline why they expect the RehaCom software to lead to better results.   

In the methods, please describe the procedure of allocation to the intervention and the control group. 

Statistical analyses: To capture the changes in cognition, you will have to perform at least a repeated measures ANOVA (mixed effects models would be better). Further, it would be helpful to adjust for covariates (age, gender, education) in the analyses.

The discussion should then evaluate why one software is better than the control training, as this is what the results report on. The authors have included those thoughts in the discussion but it is not obvious enough for the reader.

Author Response

thanks to the reviewer 1 for his/her comments.

Below our explanations:

The analysis compare the RehaCom software with a control training. The introduction must therefore emphasize that they are planning to compare the effectiveness of one training over the other. Further, it would be good if the authors outline why they expect the RehaCom software to lead to better results.  

Done. We change the title according and emphasized the training group results in all the MS.

In the methods, please describe the procedure of allocation to the intervention and the control group.

Done.

Statistical analyses: To capture the changes in cognition, you will have to perform at least a repeated measures ANOVA (mixed effects models would be better). Further, it would be helpful to adjust for covariates (age, gender, education) in the analyses.

Done. We employed a repeated measures ANOVA for fixed effects, adjusted for covariates age, schooling and time of evolution, with new results.

The discussion should then evaluate why one software is better than the control training, as this is what the results report on. The authors have included those thoughts in the discussion but it is not obvious enough for the reader.

Done. We reorganized the discussion and explained separately the advantages of the computer training.

Reviewer 2 Report

This is a manuscript describing a  well-organized project using a two-group comparison study design. The is a solid background and logical argument presented in support of the methods used along with further discussion of its findings to add clinical relevance to study results. I have provided detailed comments and suggestions in the attached PDF document to make this an even stronger submission.

Author Response

Answers to the reviewer 2:

We really appreciate the comments and suggestions of the reviewer. We uploaded a new manuscript with new material and the changes suggested.

We introduce the following changes:

A more specific title: “Effectiveness of a computer-based training program of attention and memory in patients with acquired brain damage”

Introduction was modified.

a)     A definition of acquired brain damage was included

b)     the references related with animal models (enrichment environment) in TBI and metacognition were eliminated because there were not relevant to the objective of this study

c)     the objectives were more explicit.  

Material and Methods

a)     we reformulate the sample description with more information about the recruitment and sampling procedures

b)    we included a caption in the table 1 to explain SEM, time of evolution and educational level concepts

c)     Clarification of the pre and post neuropsychological assessment using pencil and paper

d)    Substitution of the term “trials” for training sessions

e)     Explanation about the feedback to the patients

Discussion

a)     We reorganized the discussion eliminating the repetition.

My best regards

Maria L. Bringas
